# Serology suggests adequate safety measures to protect healthcare workers from COVID-19 in Shiga Prefecture, Japan

Tokuhiro Chano[1]*, Shin-ya Morita[2], Tomoyuki Suzuki[3], Tomoko Yamashita[1], Hirokazu Fujimura[1], Tatsushi Yuri[2], Masakazu Menju[1], Masaaki Tanaka[1], Fumihiko Kakuno[3]

1 Department of Clinical Laboratory Medicine, Shiga University of Medical Science Hospital, Otsu, Shiga, Japan, 2 Department of Pharmacy, Shiga University of Medical Science Hospital, Otsu, Shiga, Japan, 3 Shiga Prefecture Administration, Otsu, Shiga, Japan

* chano@belle.shiga-med.ac.jp

**Data Availability Statement:** All relevant data are within the article and its Supporting Information files. A pre-print of this article is available at DOI 10.2139/ssrn.3893783.

## Abstract

Healthcare workers (HCWs), especially frontline workers against coronavirus disease 2019 (COVID-19), are considered to be risky because of occupational exposure to infected patients. This study evaluated the correlation between seroprevalence of severe acute respiratory syndrome coronavirus 2 (SARS-CoV-2) antibodies among HCWs and the implementation of personal protective equipment (PPE) & infection prevention and control (IPC). We recruited 1237 HCWs from nine public COVID-19-designated hospitals in Shiga Prefecture, central Japan, between 15–26 February 2021. All participants answered a self-administered questionnaire and provided blood samples to evaluate SARS-CoV-2 antibodies. A total of 22 cases (1·78%) were seropositive among the 1237 study participants. An unavoidable outbreak of SARS-CoV-2 had occurred at the terminal care unit of one hospital, before identifying and securely isolating this cluster of cases. Excluding with this cluster, 0·68% of HCWs were suspected to have had previous SARS-CoV-2 infections. Binomial logistic regression from individual questionnaires and seropositivity predicted a significant correlation with N95 mask implementation under aerosol conditions ($p = 8.63e^{-06}$, aOR = 2.47) and work duration in a red zone ($p = 2.61e^{-04}$, aOR = 1.99). The institutional questionnaire suggested that IPC education was correlated with reduced seropositivity at hospitals. Seroprevalence and questionnaire analyses among HCWs indicated that secure implementation of PPE and re-education of IPC are essential to prevent SARS-CoV-2 infection within healthcare facilities. Occupational infections from SARS-CoV-2 in healthcare settings could be prevented by adhering to adequate measures and appropriate use of PPE. With these measures securely implemented, HCWs should not be considered against as significantly risky or dirty by local communities.

## Introduction

Since the start of the coronavirus disease 2019 (COVID-19) pandemic in the Shiga Prefecture, Central Japan, in April 2020, a total of 2,466 cases were recorded as of February 2021 (Fig 1).

**Funding:** Tokuhiro Chano was funded with the Grant No. PT330017, as an administrative investigation by the Shiga Prefecture Governor. Laboratory tests were partly funded from Shiga University of Medical Science Hospital budget. The funders did not have any role in the study design, data collection, data analysis, interpretation, or report writing.

**Competing interests:** The authors have declared that no competing interests exist.

The number COVID-19 cases recorded in the prefecture was about half of the mean of cases in Japan, i.e., from April 2020 to February 2021, the average number of COVID-19 cases per 100,000 population was 175 in the Shiga Prefecture and 342 nationwide [1]. During the pandemic, healthcare workers (HCWs) have been treating patients with COVID-19 in hospitals, and have restricted various aspects of their own daily lives, so as to not spread the disease from hospitals to the general population. During the several months of this pandemic, HCWs struggled to treat COVID-19 patients even with the shortage of personal protective equipment (PPE). Meanwhile, the Shiga Prefectural administration had to prepare hospitals designated specifically for COVID-19 rapidly. Regardless of such efforts, HCWs, especially frontline workers designated for COVID-19 areas, named the red zone areas, have been considered to be at increased risk of the disease, owing to their occupational exposure to infected patients. In Japan, the seroprevalence of severe acute respiratory syndrome coronavirus 2 (SARS-CoV-2) was reported to be relatively higher in HCWs [2, 3]. Because of these, the HCWs and their relatives have often been considered against as being dirty or risky according to the Japanese local communities.

In order to evaluate whether HCWs have been at increased risk of COVID-19, how the PPE usage and infection prevention and control (IPC) guidelines were invaded by SARS-CoV-2 in healthcare facilities, and what should be implemented for better protection of HCWs from the infection, this survey was conducted during 15–26 February 2021, approximately 11 months

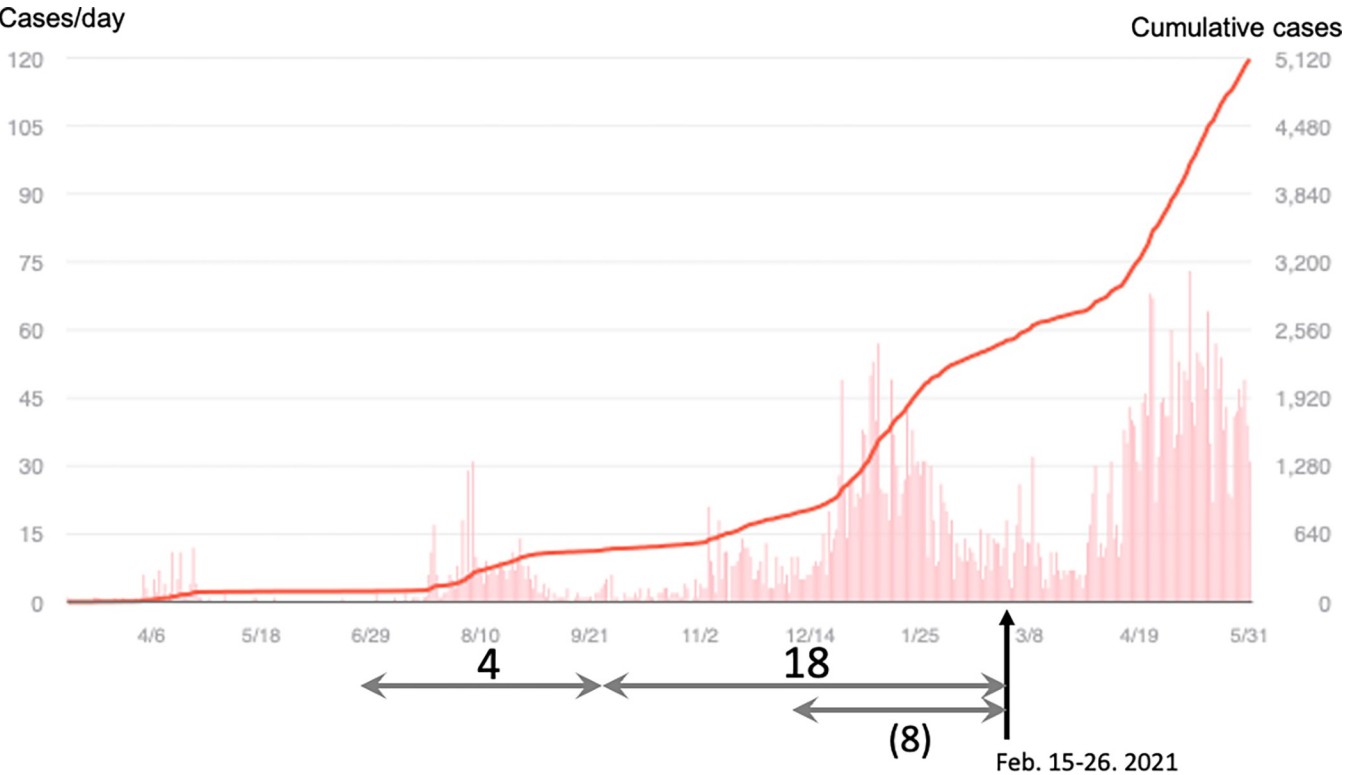

**Fig 1. Numbers of new and cumulative cases of COVID-19 in Shiga Prefecture.** New and cumulative cases of COVID-19 are indicated as bars and lines, respectively (https://covid2019.fa.xwire.jp/#japan_prefecture). The investigation and blood sampling were performed 15–26 February 2021, at the terminal timing of the third wave of the pandemic in Shiga Prefecture. The numbers and arrows below the graph indicate the estimated cases and timing of HCWs exposed to SARS-CoV-2. Four and 18 cases presumably suffered from SARS-CoV-2 at the second and third waves, respectively. Eight patients were likely exposed during a period of less than three months. *Note.* COVID-19, coronavirus disease 2019; HCW, healthcare workers; SARS-CoV-2, severe acute respiratory syndrome coronavirus 2.

after the pandemic started to affect Shiga Prefecture. Via the serological surveillance of SARS--CoV-2 antibodies among HCWs, we speculated whether the seroprevalence of HCWs was higher and whether the timing of HCWs' exposure to SARS-CoV-2 was earlier than those of the general population in Japan. Additionally, via the questionnaires' analyses from individual HCWs and representatives of healthcare institutions, we evaluated whether HCWs had securely implemented PPE and IPC.

## Methods

### Study design and participants

A cross-sectional study of HCWs was performed as a prefectural administrative investigation in nine public hospitals designated for COVID-19 in Shiga Prefecture, central Japan. Under the health & infection managements of Shiga Prefecture administration, the red & green zone management and questionnaire investigation were similarly conducted in all the participated hospitals. The investigations took place during 15–26 February 2021, which corresponded to the period between the third and fourth waves of COVID-19 in Japan. The seroprevalence of SARS-CoV-2 antibodies at the timing of this survey was estimated to be about 1–2%, and on that basis, a survey of 1000–2000 participants had been planned in order to allow evaluating at least two variables in a logistic regression analysis. Thus, this survey recruited 1237 HCWs from nine public hospitals. Participation in the study was voluntary. HCWs were invited by advertisements and/or internal announcements to participate in the study. Those interested in the study were asked to contact the study team for an appointment. If the participant had a history of working in the ward designated for COVID-19 patients, he or she was defined as a red zone worker and included in the study. However, to securely assess the risk between the red zone workers and the general (green zone) workers in the hospital workplaces, quota sampling was applied in each hospital. The ratio between the red and green zone workers was approximately 1:1.

### Data collection

Each consenting participant was given a self-administered questionnaire (S1 File) to capture the implementation of PPE, adherence with recommended IPC measures, and history of exposure to SARS-CoV-2 during the previous 11 months from April 2020 to February 2021. A representative of each institution was provided with another questionnaire (S2 File) to capture the practices of PPE usage and IPC measures in their respective hospitals.

These questionnaires originated from the protocol 'Assessment of potential risk factors or 2019-novel coronavirus (2019-nCoV) infection among HCWs in a healthcare setting', published by the World Health Organization [4] and were designed for HCWs in Japan, referring also to the checklists of Japanese society for infection prevention and control [5] and of national institute of infectious diseases [6]. From each participant, 5 mL of peripheral venous blood was collected for serological testing of SARS-CoV-2 antibodies.

### Serological tests of SARS-CoV-2 antibodies

The blood samples were separated by centrifugation, and serum was frozen until antibody evaluation. After all the study samples were collected, the serum samples were defrosted and detection of SARS-CoV-2 antibodies was conducted using the Roche Cobas® 8000 (Roche, Basel, Switzerland) and the Abbott ARCHITECT® i1000SR (Abbott, Chicago, IL) platforms at the departments of Clinical Laboratory Medicine and Pharmacy, respectively, at the Shiga University of Medical Science Hospital. Roche Cobas® was used to measure serum antibodies

specific to SARS-CoV-2 nucleocapsid (Elecsys® Anti-SARS-CoV-2 RUO), and to the receptor binding domain (RBD) of spike protein (Elecsys® Anti-SARS-CoV-2 S RUO) [7–11]. Abbott ARCHITECT® assays were used to measure specific immunoglobulin (Ig)M to spike protein (SARS-CoV-2 IgM), IgG to nucleocapsid (SARS-CoV-2 IgG), and IgG to RBD of spike protein (SARS-CoV-2 IgG II Quant) [7, 8, 11, 12]. The measured values adjusted with the manufacturers' calibrators/standards were interpreted as positives, with a cut-off index (COI) of $\geq 1 \cdot 0$, $\geq 0 \cdot 8$ U/mL, $\geq 1 \cdot 0$, $\geq 1 \cdot 4$, and $\geq 50 \cdot 0$ AU/mL, respectively. On the post-infectious Day 14, the sensitivity & specificity of each kit were 100% & 99.81%, 98.8% & 99.98%, 95% & 100%, 100% & 99.63%, and 100% & 99.9%, respectively [13, 14]. Individuals who were serologically positive for antibodies specific to the RBD of spike protein in both Elecsys® Anti-SARS-CoV-2 S RUO (Roche) and ARCHITECT® SARS-CoV-2 IgG II Quant (Abbott) were considered to have been previously infected with COVID-19 during the 11 months prior to entering the study.

## Statistical analyses

To securely assess the risk between red and green zone workers in the hospitals, chi-square or Fisher's exact tests were applied. To evaluate whether the questionnaire responses for each individual HCW were predictive of SARS-CoV-2 antibody seropositivity, analyses of binomial logistic regression following univariate correlation were performed from each question in the questionnaire to the seropositivity. Among variables, excluding those with variance inflation factor (VIF) >10 from the model due to the risk of multicollinearity, and including those indicated with p<0.05 by Fisher's exact test, binomial logistic regression was initially performed using the stepwise variable reduction method using p-Value. Individual questionnaires mainly included questions for the implementation of PPE and adherence to recommended IPC measures. In this cohort, referring to the seroprevalence and number of participants, the confirmatively regression analysis was conducted with only 2 variables, N95 mask implementation under possible aerosol conditions and working period in the red zone. To evaluate whether the questionnaires from each institute's representative were predictive of hospitals where there were seropositive HCWs, that is, where the protection barrier breakage had incidentally occurred in the hospitals designated for COVID-19, univariate and Spearman's rank correlation coefficient analyses were applied to the questions in the institutional questionnaire. The institutional questionnaire mainly asked representatives about the practical implementation of PPE and IPC educational programs. In this study, only 9 institutes were included, and both the seropositive case numbers and rates of each institute were out of normal distribution through Shapiro-Wilk normality tests (p = $6.66e^{-05}$ and $7.80e^{-05}$, respectively). Thus, each institutional difference was evaluated using the Kruskal-Wallis test followed by Holm's post-hoc significance test. Fisher's exact test and Spearman's rank correlation coefficient were also conducted to select and evaluate correlating factors in the institutional questionnaire to the seropositive hospital cases. The analysis was performed using Easy R software version $4 \cdot 1 \cdot 0$ [15].

## Ethical statement

Written informed consent were obtained from all study participants. During the process of obtaining consent, all participants were informed of the need to publish the results. All participants' personal information was anonymised during the write-up, and no participant is identifiable in the publication. This study was approved as an administrative investigation of Shiga Prefecture by the Research Review Board of Shiga University of Medical Science (No. RRB20-032). Findings from the study have been disseminated to study participants, HCWs, through a newsletter and each institute representative that represent HCWs' communities.

## Results

### Seroprevalence of total antibodies against SARS-CoV-2 and the presumed exposure history

The study population of 1237 HCWs comprised 257 (20·8%) medical doctors, 817 (66·0%) nurses, 67 (5·4%) office workers, and 96 others (7·8%) across the nine study hospitals. The other characteristics of the present study participants couldn't be precisely clarified, because individual questionnaire didn't include such questions. However, the second serological survey of HCWs was similarly conducted December 2021 in Shiga prefecture. In the second survey, 1600 HCWs was characterized with age of 42.1 ±16.4 (indicating mean ±S.D. below) years old, body mass index of 23.4 ±8.7 kg/m$^2$, and 66.2% of female predominance. Thus, the present study was presumably composed with similar character population.

A total of 22 out of 1237 HCW samples were categorised as serologically positive, meaning that 22 HCWs had been previously infected with COVID-19 during the previous 11 months. Although IgM specific to the spike protein can only be detected 2–3 months post infection, the IgG counterpart is continuously detected between 8 months and 1 year from the initial infection of SARS-CoV-2 [9, 11, 13, 16–20]. IgG specific to the SARS-CoV-2 nucleocapsid shows serological positivity for approximately 3–6 months [8, 20–22]. Using these information, we approximately estimated the timing of exposure to SARS-CoV-2 in 22 serologically positive cases. Thus, four HCWs with serological detection of only IgG specific to the spike protein were infected with COVID-19 during the second wave of the pandemic in Shiga Prefecture, and the other 18 individuals were likely infected during Shiga Prefecture's third wave (Fig 1 and S1A Fig). Eight of the 18 individuals with serologically detected IgM specific to the spike protein, in addition to the nucleocapsid-specific IgG and the spike protein-specific IgG (S1B Fig), were likely exposed to SARS-CoV-2 1–3 months prior to this investigation. The proportion of COVID-19 cases in the second and third waves in Shiga Prefecture was similar for HCWs and the general public (4 vs. 18, and 450 vs. 2011 cases, respectively; Fig 1).

### Risk comparison between red zone and green zone hospital workers

The total seroprevalence of SARS-CoV-2 antibodies was 1·78% (22/1237 cases). Twenty individuals out of 611 (3·27%) from the red zone and two out of 626 (0·32%) from the green zone were previously infected with COVID-19, with the difference between both zones being highly significant (red zone, $p$ = 0·0000485; Table 1). However, one hospital indicated the highly significant prevalence of SARS-CoV-2 positive cases in comparison with the other eight hospitals (Kruskal-Wallis test followed by Holm's post-hoc significance, $p < 0.0001$; S2 Fig). In fact, an inevitably silent outbreak of SARS-CoV-2 had occurred among patients with primary and metastatic lung cancers at the terminal care unit of one hospital. In this specific hospital, the

**Table 1. Risk comparison between red zone and green zone workers in Shiga Prefecture.**

| Working zone | | Antibody against SARS-CoV-2 | | Total | |
|---|---|---|---|---|---|
| | | Negative (-) | Positive (+) | n | p-Value |
| | Red | 591 | 20 | 611 | 4.85e-5 |
| | Green | 624 | 2 | 626 | |
| | General seroprevalence 1·78% * | | | | |
| | Red | 494 | 6 | 500 | 0.0633 |
| | Green | 524 | 1 | 525 | |
| | General seroprevalence 0·68% # | | | | |

*Note.* SARS-CoV-2, severe acute respiratory syndrome coronavirus 2; Models including (*) or excluding (#) with a hospital which the cluster happened.

HCWs could not completely implement their PPE protocols before identifying and securely isolating this cluster of cases. Therefore, there was an abnormally high prevalence of HCWs with SARS-CoV-2 antibodies at one of the study's hospitals. Thus, in evaluating the general seroprevalence of HCWs in Shiga Prefecture, we additionally considered another model excluded this specific hospital (n = 212 HCWs) and evaluated the risks of 500 red zone workers compared with those of 525 green zone workers, and found that the general seroprevalence of SARS-CoV-2 antibodies was 0·68% (7/1025 cases) in the model, and that the exposed risks of SARS-CoV-2 were not highly significant in red zone workers in Shiga Prefecture (*p* = 0·0633; Table 1).

## Secure implementation of PPE was required for protecting HCWs against COVID-19

Univariate analysis (Table 2) between SARS-CoV-2 seropositivity and individual characteristics of the self-administered questionnaire indicated possible correlations between HCWs with close contact with COVID-19 patients, close contact with the other workers in the hospital, everyday changing of the hospital uniform, working experience in the red zone section, and everyday changing of the red zone uniform. However, binomial logistic regression analysis among such significant variables could indicate only 3 factors; infected history of COVID-19 (adjusted odds ratio [aOR] 319; 95% confidence interval [CI] 22·2–4600; *p* = 0·0000227), N95 mask implementation under possible aerosol conditions (aOR 2·69; 95% CI 1·61–4·49; *p* = 0·000148), and working period in the red zone (aOR 2·06; 95% CI 1·04–4·08; *p* = 0·0377), as highly significant factors of SARS-CoV-2 seropositivity (Table 2). In this cohort, considering the seroprevalence and number of participants (1·78% and 1237 HCWs, respectively) and also the predictively important factors, we excluded the previously infected history of COVID-19, and conducted the subsequently binomial logistic regression analysis with only 2 variables, N95 mask implementation and working period in the red zone, whose aOR indicated 2·47 and 1·99 (*p* = $8·63e^{-06}$ and $2·61e^{-04}$; Table 3), respectively. In the confirmative model using only these 2 factors, these VIFs indicated 1.21 and 1.21, respectively, in which case multicollinearity was likely very little. In the model, the area under the receiver operating characteristic curve (AUC) was 0·807 (95% CI 0·707–0·907; Fig 2).

## Re-education of IPC were suggested to protect the hospital barrier against SARS-CoV-2

SARS-CoV-2 seropositive cases were detected in four of the nine hospitals investigated. These four hospitals were compared to the other five in terms of questions on the institutional questionnaire. Next, among the 54 questions for practical implementation of IPC and PPE usage, only re-education practice of IPC for HCWs suggested a highly significant correlation to protect the hospital barrier against COVID-19 (Table 4 and S3 Fig). In addition to Fisher's exact test, IPC re-education was strongly correlated with SARS-CoV-2 HCW seropositivity cases at each hospital on Spearman's rank correlation coefficient analysis (ρ = 0.949, *p* = 0.0000958; S3A Fig). Additionally, together with the numbers of both participating individuals in this survey and hospitalised COVID-19 patients in each hospital, multiple regression analysis of IPC re-education was performed to predict an association with SARS-CoV-2 seropositive cases (S3B Fig).

## Discussion

This study has indicated that red zone HCW with adequate implementation of PPE and IPC was not a highly significant risk of COVID-19, and should not have been considered against. If

**Table 2. Individual characteristics and seroprevalence of SARS-CoV-2.**

| Feature | | Antibody against SARS-CoV-2 | | | Binomial Logistic Regression | | |
|---|---|---|---|---|---|---|---|
| | | Negative (-) | Positive (+) | p-Value (Fisher's exact) | aOR | (95% CI) | p-Value |
| Red-zone working period: **Days** | | | | | | | |
| <1 week | | 395 | 1 | 3.56e-6 | **2.06** | **(1.04–4.08)** | **3.77e-2** |
| 1<2 weeks | | 272 | 0 | | | | |
| 2<3 weeks | | 98 | 6 | | | | |
| 3<4 weeks | | 81 | 4 | | | | |
| 4< weeks | | 254 | 8 | | | | |
| Previously suffered COVID-19: **Infected History** | | | | | | | |
| No infection | | 1198 | 14 | 1.24e-12 | **319** | **(22.2–4600)** | **2.27e-5** |
| Infected | | 4 | 8 | | | | |
| Close contact to patients: Pt_Contact | | | | | | | |
| No contact | | 543 | 5 | 0.0289 | 1.47 | (0.283–7.64) | 0.646 |
| Close contact | | 588 | 16 | | | | |
| N95 mask implementation under aerosol situations: **Q5** | | | | | | | |
| No situation | | 332 | 1 | 3.78e-4 | **2.69** | **(1.61–4.49)** | **1.48e-4** |
| Applied: | 1 | 716 | 12 | | | | |
| | 2 | 59 | 1 | | | | |
| | 3 | 31 | 0 | | | | |
| Non-applied: | 4 | 67 | 7 | | | | |
| Never talked with others in the hospital: Q12 | | | | | | | |
| Applied: | 1 | 1069 | 18 | 0.0451 | 0.751 | (0.268–2.10) | 0.585 |
| 2 | | 71 | 0 | | | | |
| 3 | | 37 | 3 | | | | |
| Non-applied: | 4 | 31 | 1 | | | | |
| Everyday changing the hospital uniform: Q29 | | | | | | | |
| Applied: | 1 | 371 | 6 | 0.0205 | 0.948 | (0.540–1.66) | 0.851 |
| | 2 | 205 | 9 | | | | |
| | 3 | 140 | 3 | | | | |
| Non-applied: | 4 | 422 | 3 | | | | |
| Everyday changing the red-zone uniform: Q43 | | | | | | | |
| Applied: | 1 | 403 | 8 | 1.32e-4 | 1.30 | (0.670–2.53) | 0.435 |
| | 2 | 62 | 10 | | | | |
| | 3 | 37 | 1 | | | | |
| Non-applied: | 4 | 88 | 1 | | | | |

*Note*. SARS-CoV-2, severe acute respiratory syndrome coronavirus 2; COVID-19, coronavirus disease 2019

aOR, adjusted odds ratio; CI, confidence interval.

HCWs had had an increased risk of SARS-CoV-2 exposure, they would have been affected by COVID-19 earlier or more than the general population. In fact, SARS-CoV-2 seroprevalence and the presumed timing of those infected were similar to those of the general population (Fig 1), even though the PPE shortage caused HCWs to struggle throughout several months of the pandemic in Shiga Prefecture. SARS-CoV-2 seroprevalence in HCWs in Shiga Prefecture (1·78–0·68%) was similar to that of the general population in December 2020 across various parts of Japan (Tokyo 1·35%, Aichi 0·71%, Osaka 0·69%, Fukuoka 0·42%, and Miyagi 0·14%) [23]. Theoretically calculated from each prefecture's population number and antibody prevalence, the seropositive population numbers of SARS-CoV-2 had been approximately 3–5 times

**Table 3. The subsequently binomial logistic regression analysis with 2 variables.**

| Feature | | Antibody against SARS-CoV-2 | | | Binomial Logistic Regression | | |
|---|---|---|---|---|---|---|---|
| | | Negative (-) | Positive (+) | p-Value (Fisher's exact) | aOR | (95% CI) | p-Value |
| Red-zone working period: **Days** | | | | | | | |
| <1 week | | 395 | 1 | 3.56e-6 | **1.99** | **(1.38–2.89)** | **2.61e-04** |
| 1<2 weeks | | 272 | 0 | | | | |
| 2<3 weeks | | 98 | 6 | | | | |
| 3<4 weeks | | 81 | 4 | | | | |
| 4< weeks | | 254 | 8 | | | | |
| N95 mask implementation under aerosol situations: **Q5** | | | | | | | |
| No situation | | 332 | 1 | 3.78e-4 | **2.47** | **(1.66–3.67)** | **8.63e-06** |
| Applied: | 1 | 716 | 12 | | | | |
| | 2 | 59 | 1 | | | | |
| | 3 | 31 | 0 | | | | |
| Non-applied: | 4 | 67 | 7 | | | | |

*Note.* SARS-CoV-2, severe acute respiratory syndrome coronavirus 2; COVID-19, coronavirus disease 2019

aOR, adjusted odds ratio; CI, confidence interval.

more than the number of COVID-19 cases diagnosed by polymerase chain reaction (PCR) and/or antigen tests at that time [1]. From these theoretical numbers and together with PCR and/or antigen-diagnosed COVID-19 numbers in Shiga Prefecture, we were able to calculate the seroprevalence rate of the general population of Shiga Prefecture at the time of our investigation. In doing so, a hypothetical 1–0·3% prevalence was calculated in the general population, and the seroprevalence of HCWs (1·78–0·68%) was not so higher than that of the general population in the Shiga Prefecture. In addition, HCWs' seroprevalence in the Shiga Prefecture wasn't so higher than that of another prefecture's hospital workers (1·1%) in the same time frame of February to April 2021 [24]. In Japan, previous investigations had reported that the seroprevalence was higher in HCWs [2, 3], and the data may have misled the local communities into recognizing the HCWs and their relatives as being significantly dirty or risky. However, at least in Shiga Prefecture, occupational infections from SARS-CoV-2 in healthcare settings weren't so higher than those of the generals, and we believe that HCWs should not be considered against as significantly risky or dirty by local communities.

As a matter of course, recognition of previous COVID-19 infection, that is, any symptomatic and/or diagnosed history of COVID-19 prior to the survey, indicated a maximum aOR to SARS-CoV-2 seropositivity (Table 2). However, about two-thirds of the seropositive cases (14/22 cases) could not recognise any symptoms or diagnoses in their previous histories. This reflects the difficulty of dealing with the silent invasion of SARS-CoV-2 into healthcare facilities. In all medical institutes, HCWs were checked every day for their subjective symptoms, such as fever, cough, sore throat, general malaise, dyspnoea, and abnormal taste or smell, and were under adequate IPC. However, it may be difficult to completely prevent SARS-CoV-2 invasion even in hospitals. In addition to securing IPC, if hospitals are not able to perform regular screening of HCWs using PCR and/or antigen tests, silent outbreaks of SARS-CoV-2 may incidentally occur even in healthcare facilities. In fact, among investigated hospitals, 15 cases of HCWs could not implement their PPE protocols before identifying and securely isolating SARS-CoV-2 cases at the terminal care unit of lung cancers, and an unavoidable outbreak occurred. This conformed that the greatest risk to HCWs may be their own colleagues or patients in the early stages of unsuspected infections rather than red zone working [25]. It's

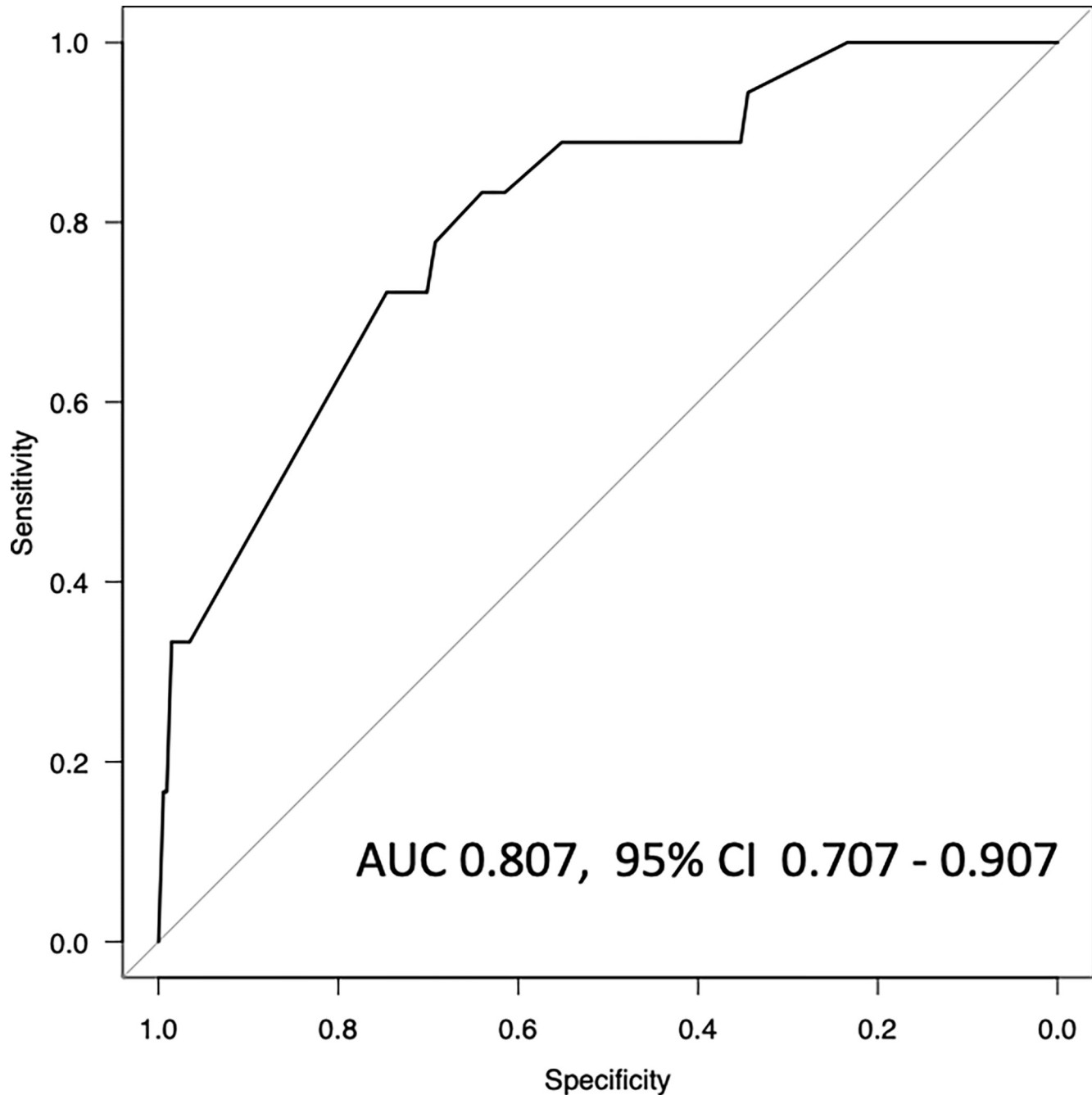

**Fig 2. Area under the receiver operating characteristic curve (AUC) for the prediction of SARS-CoV-2 seropositivity of healthcare workers.** The model was composed of only two factors of N95 mask implementation under possible aerosol conditions and working period in the hospital red zone section. AUC was 0·807 (95% CI 0·707–0·907). *Note.* AUC, area under the receiver operating characteristic curve; CI, confidence interval.

also in line with that HCWs working in general, ophthalmology, and respiratory departments were prone to risk compared with HCWs working in the infection department [26].

Apart from recognition of previous COVID-19 infections, the binomial logistic regression analysis from the individual questionnaire to SARS-CoV-2 seropositivity identified two highly significant factors: N95 mask implementation under possible aerosol conditions and working period (in days) in the red zone section. Among PPE, N95 mask implementation should be

**Table 4. Institutional comparison for SARS-CoV-2 seropositivity of hospital.**

| Feature | Seropositivity of hospital | | Fisher's exact test |
|---|---|---|---|
| | Negative (-) | Positive (+) | p-Value |
| IPC re-education: Q50 | | | |
| Performed (Yes) | 5 | 0 | 7.94e-3 |
| Not (No) | 0 | 4 | |

*Note*. SARS-CoV-2, severe acute respiratory syndrome coronavirus 2; IPC, infection prevention and control.

especially required under possible aerosol conditions in hospitals. Even if red zone HCWs could be safely protected with PPE and IPC, the working duration in the red zone should be as distributed as far as possible. This is comparable hat more prolonged contact with COVID-19 patients remains a crucial risk factor for SARS-CoV-2 [27]. In order to reduce the opportunities of SARS-CoV-2 exposure, we should shorten the working periods in the red zone for HCWs, as far as possible. The efforts for two highly significant aspects should be continuously performed in healthcare facilities.

The analysis from the institutional questionnaire suggested a correlation between practical re-education of IPC for HCWs and SARS-CoV-2 seropositive hospital cases. There were few participating hospitals in this investigation, and we could not indicate the absolute significance of IPC re-education, which is possibly most important for implementing IPC and PPE. If the number of individuals participating in this survey and/or hospitalised COVID-19 patients in each hospital were to increase, SARS-CoV-2 seropositive HCWs could be increased in each hospital. Therefore, we performed multiple regression analysis of IPC re-education with these numbers to predict an association with SARS-CoV-2 seropositive cases (S3B Fig). The analysis indicated a weak association with IPC re-education even without its statistical significance, so IPC re-education may still give each hospital the capacity to accept more than 100 hospitalised COVID-19 patients.

Although this study provides interesting correlation between SARS-CoV-2 seroprevalence and the implementation of PPE & IPC in HCWs, it is important to note that there are some limitations. The survey was performed before alpha-variant predominant expansion in Japan, and the results might not reflect that in the predominant period of alpha, delta, or omicron variant, etc. Nevertheless, the implementation procedure of PPE and IPC is likely similar to that of the original SARS-CoV-2, so the study data will contribute to the preventive measures of the present and future variants in healthcare facilities. Meanwhile, to evaluate the statistical correlation to multi-variables, the sample size of this study might not be enough. The cross-sectional nature of this study also doesn't allow to obtain conclusive causal evidence. Thus, referring to the seroprevalence and number of participants, we have focused on 2 factors, N95 mask implementation under possible aerosol conditions and working period in the red zone, in this correlation analysis of Shiga Prefecture, central Japan. The occupational infection of SARS-CoV-2 in healthcare settings could be prevented by adherence to adequate measures and appropriate use of PPE like N95 mask. Still, the working period in the red zone should be as distributed for HCWs as far as possible, to reduce the occupational opportunities of SARS-CoV-2 exposure. The analysis also indicated an association between SARS-CoV-2 seropositivity and IPC re-education, so practical re-education of IPC for HCWs may contribute to accept hospitalised COVID-19 patients into hospitals. The study findings should be confirmed with more cases of participants, and PPE & IPC strategy should be adapted to predominant variants of SARS-CoV-2 in future. Nevertheless, this survey indicated that red zone HCWs with adequate implementation of PPE and IPC were not at high risk of SARS-CoV-2 exposure, and HCWs should not be considered against as being risky or dirty by local Japanese communities.

## Conclusions

This study indicated that secure implementation of PPE and re-education of IPC were essential to prevent SARS-CoV-2 infection within healthcare facilities. HCWs with adequate implementation of PPE and IPC should not be considered against as significantly risky or dirty by local communities.

## Supporting information

**S1 Fig. Serological values of SARS-CoV-2 antibodies in 22 positive cases.**
(TIFF)

**S2 Fig. Comparison for SARS-CoV-2 seropositivity of healthcare workers among nine hospitals designated for COVID-19 in Shiga Prefecture.**
(TIFF)

**S3 Fig. Infection prevention control re-education was likely correlated with SARS-CoV-2 seropositivity of healthcare workers at each hospital.**
(TIFF)

**S1 File. Individual questionnaire.**
(XLSX)

**S2 File. Institutional questionnaire.**
(XLSX)

**S3 File. Individual data in Shiga Prefecture.**
(XLSX)

**S4 File. Institute data in Shiga Prefecture.**
(XLSX)

## Acknowledgments

We are grateful to Hiroko Kita and Saeko Katsumoto for inputting the data from the questionnaires. We appreciate the directors of the nine hospitals who participated in this study for their support throughout the study. We would also like to thank Shiga Prefecture Governor, Taizo Mikazuki, for his funding decision and permission to publish this study. Finally, we would like to express our deepest condolences to a colleague, Masaaki Tanaka, who died during this survey.

## Author Contributions

**Conceptualization:** Tokuhiro Chano, Tomoyuki Suzuki.

**Data curation:** Tokuhiro Chano, Shin-ya Morita, Tomoyuki Suzuki, Tomoko Yamashita, Hirokazu Fujimura, Tatsushi Yuri.

**Formal analysis:** Tokuhiro Chano, Tomoyuki Suzuki.

**Funding acquisition:** Tokuhiro Chano.

**Investigation:** Tokuhiro Chano, Tomoyuki Suzuki.

**Methodology:** Tokuhiro Chano, Shin-ya Morita, Tomoyuki Suzuki, Tomoko Yamashita, Hirokazu Fujimura, Tatsushi Yuri.

**Project administration:** Tokuhiro Chano, Tomoyuki Suzuki, Fumihiko Kakuno.

**Resources:** Tokuhiro Chano, Tomoyuki Suzuki, Fumihiko Kakuno.

**Supervision:** Masakazu Menju, Masaaki Tanaka, Fumihiko Kakuno.

**Validation:** Tokuhiro Chano, Tomoyuki Suzuki.

**Visualization:** Tokuhiro Chano.

**Writing – original draft:** Tokuhiro Chano, Shin-ya Morita, Tomoyuki Suzuki.

**Writing – review & editing:** Tokuhiro Chano, Tomoyuki Suzuki, Masakazu Menju, Masaaki Tanaka, Fumihiko Kakuno.

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
