## [Decision Letter · Decision Letter 0]

5 Jan 2022

PONE-D-21-35904Serology suggests adequate safety measures to protect healthcare workers from COVID-19 in Shiga Prefecture, JapanPLOS ONE

Dear Dr. Chano,

Thank you for submitting your manuscript to PLOS ONE. After careful consideration, we feel that it has merit but does not fully meet PLOS ONE’s publication criteria as it currently stands. Therefore, we invite you to submit a revised version of the manuscript that addresses the points raised during the review process.

We look forward to receiving your revised manuscript.

Kind regards,

Gabriel O Dida, PhD

Academic Editor

PLOS ONE

3. Thank you for stating the following in the Funding and Acknowledgments Section of your manuscript:

“Funding:

The study cost was mainly funded as an administrative investigation by the Shiga Prefecture Governor. Laboratory tests were partly funded from Shiga University of Medical Science Hospital budget. The fund numbers are not applicable. The funders did not have any role in the study design, data collection, data analysis, interpretation, or report writing.

Acknowledgement:

We would also like to thank Shiga Prefecture Governor, Taizo Mikazuki, for his funding decision and permission to publish this study”

We note that you have provided information within the Funding and Acknowledgements Section. Please note that funding information should not appear in the Acknowledgments section or other areas of your manuscript. We will only publish funding information present in the Funding Statement section of the online submission form.

“No - The study cost was mainly funded as an administrative investigation by the Shiga Prefecture Governor. Laboratory tests were partly funded from Shiga University of Medical Science Hospital budget. The fund numbers are not applicable. The funders did not have any role in the study design, data collection, data analysis, interpretation, or report writing.”

Reviewers' comments:

Reviewer's Responses to Questions

**Comments to the Author**

1. Is the manuscript technically sound, and do the data support the conclusions?

Reviewer #1: Partly

Reviewer #2: Partly

Reviewer #3: Yes

2. Has the statistical analysis been performed appropriately and rigorously? 

Reviewer #1: No

Reviewer #2: No

Reviewer #3: No

3. Have the authors made all data underlying the findings in their manuscript fully available?

Reviewer #1: Yes

Reviewer #2: No

Reviewer #3: Yes

4. Is the manuscript presented in an intelligible fashion and written in standard English?

Reviewer #1: Yes

Reviewer #2: No

Reviewer #3: Yes

5. Review Comments to the Author

Reviewer #1: General comments

Dear authors, thank you for the opportunity to read your work. This paper investigates the possibility that HCWs are a source of risk for the population given their constant exposure to COVID-19 patients. Preventive safety measures are also investigated. The survey was conducted between 15 and 26 February 2021 in Shiga, Japan. Questionnaires, blood sample collection and regressive and correlational analyzes were used to find a conclusion. However, at present, I believe that there are some critical issues to be addressed.

Major comments

1) Abstract. “From these, binomial logistic regression from individual questionnaires and seropositivity predicted a significant correlation with N95 mask implementation under aerosol conditions and work duration in a red zone.” Please, specify P-value and the regression coefficient.

2) Methods.

2.1. “Binomial logistic regression analysis was applied from the individual questionnaire to SARS-CoV-2 seropositivity [...]” The use of logistic regression models requires the verification of some assumptions (https://www.statisticssolutions.com/free-resources/directory-of-statistical-analyses/assumptions-of-logistic-regression/, https://pubmed.ncbi.nlm.nih.gov/21996075/#:~:text=Basic%20assumptions%20that%20must%20be,lack%20of%20strongly%20influential%20outliers). Please, detail in this section how these have been verified.

2.2. “Each institutional difference was evaluated using the Kruskal-Wallis test followed by Holm’s post-hoc significance test.” and “[...] Fisher’s exact test and Spearman’s rank correlation coefficient were conducted to select and evaluate correlating factors in the institutional questionnaire to the seropositive hospital cases.” Generally, the use of non-parametric measures is appropriate when the dataset is not normally distributed (In some cases, some authors support the adoption of parametric tests even for non-normal datasets, https://www.ncbi.nlm.nih.gov/pmc/articles/PMC3445820/). On the other hand, other authors point out that the difference in power between parametric and non-parametric tests is small (e.g., https://www.ncbi.nlm.nih.gov/pmc/articles/PMC2743502/). Other approaches have also been proposed (e.g., https://www.ncbi.nlm.nih.gov/pmc/articles/PMC1310536/). Therefore, I suggest that the authors briefly address this aspect in the manuscript, motivating their choices to use non-parametric tests. In this regard, the verification of the shape of the datasets through tests for normality (e.g., Shapiro-Wilk + Q-Q plots) could be conclusive.

2.3. Dear authors, I ask if the possibility to test for internal consistency of questionnaires has been evaluated.

3) Results.

3.1. P-values should be used, at best, as graded measures of the strength of evidence against the null hypothesis (https://pubmed.ncbi.nlm.nih.gov/28698825/, https://www.ncbi.nlm.nih.gov/pmc/articles/PMC4877414/). Therefore, the adoption of a simple threshold can be misleading. Hence, I suggest speaking of "more and less significant" or "high and low significance" rather than "significant and non-significant."

3.2. Since P-values measure the statistical significance but not the effect size, I suggest commenting on the results also based on the intensity of the phenomenon (e.g., strong, moderate, weak associations, etc.)

4) Discussion.

4.1. I suggest making a comparison with the literature published on this topic (also relating to other countries). This serves to contextualize and better understand the relevance of the results found (e.g., https://pubmed.ncbi.nlm.nih.gov/33115772/, https://pubmed.ncbi.nlm.nih.gov/33003634/, https://pubmed.ncbi.nlm.nih.gov/33140084/).

4.2. I suggest clearly specifying the limitations of the study. In particular, i) the sample size prevents these results from being generalizable, and ii) the fundamentally cross-sectional nature of the study and the search for correlations do not allow to obtain conclusive causal evidence.

Minor comments

m1) References. Regarding websites, I suggest specifying the name of the source and providing an access date.

Reviewer #2: Reviewer’s report

Title: Serology suggests adequate safety measures to protect healthcare workers from

COVID-19 in Shiga Prefecture, Japan

Version: Date: 9th December 2021

Reviewer: Samuel Bosomprah

Reviewer’s report

The authors stated that this was a seroprevalence study aimed “to estimate the timing of HCWs’ exposure to SARS-CoV-2 and to analyse whether the HCWs had been exposed to SARSCoV-2 earlier than the general population as well as to determine whether the seroprevalence of HCWs was higher than that of the general population in Japan”

General comment:

The manuscript did not read well as a research document. The language is not consistent with a research paper write up. For example, the authors used terms such as “noted” in the introduction instead of, say, cases were “recorded” etc. Another term was “clarify” in method section “To clarify whether the questionnaire responses for each individual HCW were predictive of SARS-CoV-2 antibody seropositivity…” and “To clarify whether the questionnaires from each institute’s representative were predictive of hospitals where there were seropositive HCWs…” These are not appropriate statistical terminologies.

Introduction

• I would have expected the authors to begin the introduction with clear statement of the problem before elaborating on it in terms of the magnitude of the problem and which group is unfairly or disproportionately affected.

• The authors did not provide compelling argument for why the study is important (i.e., justification/significance/rationale) and what the knowledge/research gap is, which they are seeking to fill in

Methods

• The method section is not well structured. The authors should consider renaming “study setting” as “Study design and participants” and integrate the “study participants” texts.

• Authors should avoid the phrase “…investigation prospectively recruited..” for a cross-sectional study design

• Authors should consider integrating the subheadings “Evaluation of individual personal protective equipment (PPE)” and “Evaluation for institutional infection prevention and control (IPC)” into “Data analysis” and edit the language to read well.

• Authors did not provide the assumptions on arriving at a sample size of 1237. They should consider including a section on “sample size consideration”, which should describe how the sample size of 1237 was arrived at.

• The section “Patient and public involvement” should be renamed as “Ethical statement” and should be edited accordingly.

Findings

• “Findings” should be edited to “Results”

• The analysis and presentation of results did not meet the statistical rigour required of a seroprevalence study. The tables were poorly presented – only frequencies were presented; no corresponding proportions (seroprevalence) were presented (See tables 1, 2 and 3). P-values were inappropriately presented. It is recommended to present p-values to 3 decimal places.

Discussion

• The authors were inappropriately referencing tables and figure as if they are writing up results

• The authors should focus on interpreting the results instead of rehashing same in the discussion.

• Should discuss the strength and limitations of the study.

• They should also discuss the impact of the findings as well as prescription of future work.

• No apparent conclusion

Reviewer #3: - Despite the fact that the blood sample was collected in prospective manner, the seropositivity/ seroprevalence detection is actually (or at least what it is believed to) a measure of past historical record of covid exposure. In the absence of any baseline blood samples, a strong assumption that exposure was solely determined by the seropositivity is needed. This along with its justification should be made clear in the article. The interpretation of such data should be done carefully. Do we have any supporting evidence on how the seropositivity behaves over time? Can exposure to covid19 be perfectly covered by the seropositivity?

- Related to the above question, what is the sensitivity and specificity of the kit used wrt to past covid exposure? This should be explained in the article.

- Covid19 may have a big impact on how the disease spreads and hence in determining the green or red area. These variants are also time dependent. Can this be taken into account in the current model? If not, how do you explain the validity of the model when lacking such data?

- Removal of the terminal care unit of one hospital in which the outbreak took place from the analysis is problematic and potentially raises some questions on the definition used to determine what is the red zone in this research. If the removal is done post-hoc, i.e. upon collecting and inspecting the data, this may potentially introduce some biases at the very least. If an outbreak disqualified a red zone hospital from analysis, what will constitute a red-zone? Has this disqualification of data being defined in the research protocol or SAP prior to conducting the analysis?

- Secure implementation of PPE was required …, Pg 10 and 11, and table 1: it is not clear whether the fitted binomial logistic regression analysis was a multivariable model? If so this should be made clear and should be explained what method was used to retain those 3 variables. Was it simply variables retained in the model which had significant p-values? Was there kind or model selection such as backward, forward or stepwise attempted?

- The article should highlight some restrictions of the study as highlighted above. This should be done in the discussion.

6. PLOS authors have the option to publish the peer review history of their article (what does this mean?). If published, this will include your full peer review and any attached files.

Reviewer #1: **Yes: **Alessandro Rovetta

Reviewer #2: No

Reviewer #3: No

---

## [Author Response · Author response to Decision Letter 0]

5 Apr 2022

Response to Reviewers

Reviewer #1

Major comments

1) Abstract. 

Please specify P-value and the regression coefficient.

Following your suggestion, we’ve added p- & beta-Values of each variable, additionally into Table 2-3.

2) Methods.

2.1. The use of logistic regression models requires the verification of some assumptions.

As you have suggested, if the seroprevalence has been 1-2% and we've analyzed 8 variables, the logistic regression will require about 4000-8000 cases. Thus, binomial logistic regression was initially performed using the stepwise variable reduction method using p-Value. However, in this cohort, referring to the seroprevalence and number of participants (1.78% and 1237 cases), the confirmatively regression analysis was conducted with only 2 variables, N95 mask implementation and working period in the red zone (Page 7, Line 137 - 141).

This study limitation has also been described in the revised manuscript (Page 18, Line 371 - 382).

2.2. I suggest that the authors briefly address this aspect in the manuscript, motivating their choices to use non-parametric tests.

Only 9 institutes were included in this study, meaning small number. In addition, both the seropositive case numbers and rates of each institute were out of normal distribution through Shapiro-Wilk normality test (p=0.00006657 and 0.00007803), respectively. Thus, in order to evaluate the difference and questionnaire of each institution, we applied nonparametric measures such as the Kruskal-Wallis test and Spearman’s rank correlation coefficient, respectively, to the analyses.

These are described in the revised manuscript (Page 7 - 8, Line 141 - 147).

2.3. Dear authors, I ask if the possibility to test for internal consistency of questionnaires has been evaluated. 

Under the health & infection management of Shiga prefecture administration, the zone management and questionnaire investigation were similarly conducted in all the participated hospitals (Page 4, Line 80 - 83).

In addition, these questionnaires were designed, referring also to the checklists of Japanese society for infection prevention and control and of national institute of infectious diseases (Page 5, Line 99 - 101). 

3) Results.

3.1. I suggest speaking of "more and less significant" or "high and low significance" rather than "significant and non-significant."

3.2. I suggest commenting on the results also based on the intensity of the phenomenon.

Following your suggestion, we have modified our descriptions into "highly significant" and "weak association," etc.

4) Discussion. 

4.1. I suggest making a comparison with the literature published on this topic.

Based on your suggestion, we’ve compared with the other literatures and produced more fruitful discussion (Page 16 -17, Line 344 - 350 and 357 - 358).

4.2. I suggest clearly specifying the limitations of the study.

According to your suggestion, we have described some limitations of this study, and produced a more balanced work (Page 18, Line 371 - 389).

Minor comments

m1) References. Regarding websites, I suggest specifying the name of the source and providing an access date.

According to your suggestion, we have added the information into References.

 

Reviewer #2

General comment: The language is not consistent with a research paper write up.

Base on your suggestion, we have used adequate terminologies in the sections of “Introduction” and “Methods”.

Introduction

• The authors to begin the introduction with clear statement of the problem.

• The authors did not provide compelling argument for why the study is important

Considering your suggestion, we have reconstructed the “Introduction” section better.

Methods

• “Study design and participants”

• Authors should avoid the phrase “…investigation prospectively recruited...”

• “Data analysis”

• “Ethical statement”

Following your suggestion, we have reconstructed the “Methods” section better.

• Authors did not provide the assumptions on arriving at a sample size of 1237.

We appreciate your meaningful suggestion. As you have suggested, if the seroprevalence has been 1-2% and we've analyzed 8 variables, the logistic regression will require about 4000-8000 cases. Thus, binomial logistic regression was initially performed using the stepwise variable reduction method using p-Value. However, in this cohort, referring to the seroprevalence and number of participants (1.78% and 1237 cases), the confirmatively regression analysis was conducted with only 2 variables, N95 mask implementation and working period in the red zone (Page 7, Line 137 - 141).

In addition, the study limitation has been described in the revised manuscript (Page 18, Line 371 - 382).

Findings

• should be edited to “Results”

• Tables were poorly presented – only frequencies were presented; no corresponding proportions (seroprevalence) were presented (See tables 1, 2 and 3). P-values were inappropriately presented. It is recommended to present p-values to 3 decimal places.

We appreciate your meaningful suggestion, and have reconstructed all the tables.

Discussion

• The authors were inappropriately referencing tables and figure as if they are writing up results

• The authors should focus on interpreting the results instead of rehashing same in the discussion.

• Should discuss the strength and limitations of the study. 

• They should also discuss the impact of the findings as well as prescription of future work.

• No apparent conclusion

Base on your fruitful suggestions, we have reconstructed the “Discussion” section better. We have also described the study limitations and prescription in future (Page 18, Line 371 - 382 and 387 - 389). 

We’ve added apparently the “Conclusion” section in the revised manuscript.

 

Reviewer #3

- What is the sensitivity and specificity of the kit used to past covid exposure? 

In this context, referring to the manufacturer's’ specification documents and previously reports, we have clarified them in the “Method” section (Page 6, Line 116 - 118). 

- Do we have any supporting evidence on how the seropositivity behaves over time? Can exposure to covid19 be perfectly covered by the seropositivity?

- How the seropositivity behaves over time? Can exposure to covid19 be perfectly covered by the seropositivity?

In a healthy adult population (, not in children), the seropositivity has almost completely reflected the previous infections. Additionally, the time courses of each antibody titer decay have been in line with the previous reports. In the revised version, we have described them in the “Result” section (Page 9, Line 164 - 167).

- Covid19 variants are also time dependent. Can this be taken into account in the current model? If not, how do you explain the validity of the model when lacking such data? 

The survey was performed before alpha-variant predominant expansion in Japan, and the results might not reflect that in the predominant period of alpha, delta, or Omicron variant, etc. Nevertheless, the implementation procedure of PPE and IPC is likely similar to that of the original SARS-CoV-2, so the study data will contribute to the preventive measures of the present and future variants in healthcare facilities.

We have described this issue in the study limitations (Page 18, Line 371 - 382).

- Removal of the terminal care unit of one hospital in which the outbreak took place from the analysis is problematic. 

Following your suggestion, we have evaluated even both models included with or without the specific hospital where the outbreak took place (Table 1; Page 10, Line 194 - 205).

- It is not clear whether the fitted binomial logistic regression analysis was a multivariable model? If so this should be made clear and should be explained what method was used to retain those 3 variables. Was it simply variables retained in the model which had significant p- values? Was there kind or model selection such as backward, forward or stepwise attempted? 

As you suggested, we clarified that binomial logistic regression was initially performed using the stepwise variable reduction method using p-Value. In this cohort, referring to the seroprevalence and number of participants (1.78% and 1237 cases), the confirmatively regression analysis was conducted with only 2 variables, N95 mask implementation and working period in the red zone. 

These are described in the "Statistical analysis” of the “Method” section (Page 7, Line 137 - 141). 

- The article should highlight some restrictions of the study as highlighted above. This should be done in the discussion. 

As you suggested, we have highlighted the study limitations in the “Discussion” section (Page 18, Line 371 - 382).

---

## [Decision Letter · Decision Letter 1]

10 May 2022

PONE-D-21-35904R1Serology suggests adequate safety measures to protect healthcare workers from COVID-19 in Shiga Prefecture, JapanPLOS ONE

Dear Dr. Chano,

Thank you for submitting your manuscript to PLOS ONE. After careful consideration, we feel that it has merit but does not fully meet PLOS ONE’s publication criteria as it currently stands. Therefore, we invite you to submit a revised version of the manuscript that addresses the points raised during the review process.

The reviewers have provided additional information to help you further improve your manuscript before consideration for publication in PLOS ONE journal. 

We look forward to receiving your revised manuscript.

Kind regards,

Gabriel O Dida, PhD

Academic Editor

PLOS ONE

Journal Requirements:

Please reconsider the use of the term "discrimination" and the presentation of your conclusions. This term implies the unjust or prejudicial treatment, and your conclusions appear to suggest (by inverse of their arguments) that it would be appropriate to subject healthcare workers to such prejudicial treatment had occupational infections been detected.

Reviewers' comments:

Reviewer's Responses to Questions

**Comments to the Author**

1. If the authors have adequately addressed your comments raised in a previous round of review and you feel that this manuscript is now acceptable for publication, you may indicate that here to bypass the “Comments to the Author” section, enter your conflict of interest statement in the “Confidential to Editor” section, and submit your "Accept" recommendation.

Reviewer #1: (No Response)

Reviewer #2: (No Response)

Reviewer #3: All comments have been addressed

2. Is the manuscript technically sound, and do the data support the conclusions?

Reviewer #1: Partly

Reviewer #2: Partly

Reviewer #3: Yes

3. Has the statistical analysis been performed appropriately and rigorously? 

Reviewer #1: No

Reviewer #2: No

Reviewer #3: Yes

4. Have the authors made all data underlying the findings in their manuscript fully available?

Reviewer #1: Yes

Reviewer #2: Yes

Reviewer #3: Yes

5. Is the manuscript presented in an intelligible fashion and written in standard English?

Reviewer #1: Yes

Reviewer #2: No

Reviewer #3: Yes

6. Review Comments to the Author

Reviewer #1: Dear authors, thank you for your revisions and responses. Almost all points have been adequately addressed. However, I suggest one final important check.

1) Logistic regression assumptions are still not discussed, e.g., little or no multicollinearity, linearity of independent variables and log odds, and others (please, see the references I suggested in the previous round). This aspect is essential for validating the statistical results and should be added to the publication (this could be done in a supplementary file as well). In particular, it is necessary to test all the assumptions and report the results to consent the reader to interpret them independently. Thank you.

Reviewer #2: Reviewer’s report

Title: Serology suggests adequate safety measures to protect healthcare workers from

COVID-19 in Shiga Prefecture, Japan

Version 2: Date: 30th April 2022

Reviewer: Samuel Bosomprah

General comment:

The authors addressed some of my earlier comments. But there remains some major work especially presentation of results and tables. The specifics are stated below

Methods

• Authors should state the eligibility (inclusion and exclusion) criteria under the subsection “Study design and participants”

• “Data analyses” and Statistical analysis” sections should be integrated into one subheading called “Statistical analysis”.

• Authors did not provide the assumptions on arriving at a sample size of 1237. They should consider including a section on “sample size consideration”, which should describe how the sample size of 1237 was arrived at.

Results

• The results section should start with the subheading “Characteristics of study participants”. This section should have a table and comments describing who these participants were: example, distribution of sex, age, and other sociodemographic characteristics…

• The analysis and presentation of results did not meet the statistical rigour required of a seroprevalence study. The tables were poorly presented. P-values were inappropriately presented. It is recommended to present p-values to 3 decimal places.

• The betas in the logistic regression table are uninterpretable. Odds ratio would have been better…

Discretionary Revisions: Reject

Level of interest: An article of importance in its field

Quality of written English: Authors should proofread and correct for grammatical errors and use appropriate research language

Statistical review: No, I am a statistician and have reviewed the statistical methods used.

Declaration of competing interests: I declare that I have no competing interest

Reviewer #3: (No Response)

7. PLOS authors have the option to publish the peer review history of their article (what does this mean?). If published, this will include your full peer review and any attached files.

Reviewer #1: **Yes: **Alessandro Rovetta

Reviewer #2: No

Reviewer #3: No

---

## [Author Response · Author response to Decision Letter 1]

27 May 2022

For Academic Editor

• Please reconsider the use of the term "discrimination" and the presentation of your conclusions.

According to your suggestion, we've changed the word “discriminate” to “consider” or “recognize”.

Response to Reviewers

Reviewer #1

1) Logistic regression assumptions are still not discussed; multicollinearity should be added to the publication.

We’ve taken your advice and clarified variance inflation factor (VIF) of each variable has been used for evaluating the risk of multicollinearity in "Statistical analyses" section (Page 7, Line 132-133). 

Additionally, in the binomial logistic regression model with only 2 variables, N95 mask implementation and working period in the red zone, we’ve clarified those VIFs (Page 12, Line 233-234).

Meanwhile, also referring the suggestion of Reviewer 2, in the re-revise version of our manuscript, Tables 2-3 have been re-summarized with odds ratio rather than beta, etc.

 

Reviewer #2

Methods

• Authors should state the eligibility (inclusion and exclusion) criteria under the subsection “Study design and participants”.

We’ve taken your advice and added the rationale for the provision for red zone workers into the "Study design and participants" section (Page 5, Line 91-93).

• “Data analyses” and Statistical analysis” sections should be integrated into one subheading called “Statistical analysis”.

Following your suggestion, we have integrated these two into one section " Statistical analysis".

• Authors should provide the assumptions on arriving at a sample size of 1237. 

We’ve taken your advice and added the rationale for the size of the recruited participants into the "Study design and participants" section (Page 5, Line 85-88).

Results

• The results section should start with the subheading “Characteristics of study participants”. This section should have a table and comments describing who these participants were: example, distribution of sex, age, and other sociodemographic characteristics…

In fact, the other characteristics of the present study participants couldn't be precisely clarified, because individual questionnaire didn't include such questions. However, the second serological survey of HCWs was similarly conducted December 2021 in Shiga prefecture. In the second survey, 1600 HCWs was characterized with age of 42.1 ±16.4 (indicating mean ±S.D. below) years old, body mass index of 23.4 ±8.7 kg/m2, and 66.2% of female predominance. Thus, the present study was presumably composed with similar character population.

These have been described in the re-revised manuscript (Page 8, Line 164 – Page 9, Line 170).

• The analysis and presentation of results did not meet the statistical rigor required of a seroprevalence study. The tables were poorly presented. P-values were inappropriately presented. It is recommended to present p-values to 3 decimal places.

• The betas in the logistic regression table are uninterpretable. Odds ratio would have been better…

According to your suggestions, we have summarized and refined Tables 2-3.

---

## [Decision Letter · Decision Letter 2]

9 Jun 2022

Serology suggests adequate safety measures to protect healthcare workers from COVID-19 in Shiga Prefecture, Japan

PONE-D-21-35904R2

Dear Dr. Chano,

We’re pleased to inform you that your manuscript has been judged scientifically suitable for publication and will be formally accepted for publication once it meets all outstanding technical requirements.

Kind regards,

Gabriel O Dida, PhD

Academic Editor

PLOS ONE

Additional Editor Comments (optional):

Reviewers' comments:

Reviewer's Responses to Questions

**Comments to the Author**

1. If the authors have adequately addressed your comments raised in a previous round of review and you feel that this manuscript is now acceptable for publication, you may indicate that here to bypass the “Comments to the Author” section, enter your conflict of interest statement in the “Confidential to Editor” section, and submit your "Accept" recommendation.

Reviewer #1: All comments have been addressed

Reviewer #3: All comments have been addressed

2. Is the manuscript technically sound, and do the data support the conclusions?

Reviewer #1: Yes

Reviewer #3: Yes

3. Has the statistical analysis been performed appropriately and rigorously? 

Reviewer #1: Yes

Reviewer #3: Yes

4. Have the authors made all data underlying the findings in their manuscript fully available?

Reviewer #1: Yes

Reviewer #3: Yes

5. Is the manuscript presented in an intelligible fashion and written in standard English?

Reviewer #1: Yes

Reviewer #3: Yes

6. Review Comments to the Author

Reviewer #1: Dear authors, thank you for your final revisions. I wish you the best for this and your future research.

Reviewer #3: (No Response)

7. PLOS authors have the option to publish the peer review history of their article (what does this mean?). If published, this will include your full peer review and any attached files.

Reviewer #1: **Yes: **Alessandro Rovetta

Reviewer #3: No

---

## [Editor Report · Acceptance letter]

14 Jun 2022

PONE-D-21-35904R2 

Serology suggests adequate safety measures to protect healthcare workers from COVID-19 in Shiga Prefecture, Japan 

Dear Dr. Chano:

I'm pleased to inform you that your manuscript has been deemed suitable for publication in PLOS ONE. Congratulations! Your manuscript is now with our production department. 

Kind regards, 

on behalf of

Dr. Gabriel O Dida 

Academic Editor

PLOS ONE